# Assessment of Small-Extent Forest Fires in Semi-Arid Environment in Jordan Using Sentinel-2 and Landsat Sensors Data

Bassam Qarallah, Yahia A. Othman *, Malik Al-Ajlouni, Hadeel A. Alheyari and Bara'ah A. Qoqazeh

Department of Horticulture and Crop Science, The University of Jordan, Amman 119421, Jordan
* Correspondence: ya.othman@ju.edu.jo; Tel.: +962-779695781

**Abstract:** The objective of this study was to evaluate the separability potential of Sentinel-2A (Multi-Spectral Instrument, MSI) and Landsat (Operational Land Imager, OLI and Thermal Infrared Sensor, TIRS) derived indices for detecting small-extent (<25 ha) forest fires areas and severity degrees. Three remote sensing indices [differenced Normalized Burn Ratio (dNBR), differenced Normalized Different Vegetation Index (dNDVI), and differenced surface temperature (dTST)] were used at three forest fires sites located in Northern Jordan; Ajloun (total burned area 23 ha), Dibbeen (burned area 10.5), and Sakeb (burned area 15 ha). Compared to ground reference data, Sentinel-2 MSI was able to delimit the fire perimeter more precisely than Landsat-8. The accuracy of detecting burned area (area of coincidence) in Sentinel-2 was 7%–26% higher that Landsat-8 OLI across sites. In addition, Sentinel-2 reduced the omission area by 28%–43% and the commission area by 6%–38% compared to Landsat-8 sensors. Higher accuracy in Sentinel-2 was attributed to higher spatial resolution and lower mixed pixel problem across the perimeter of burned area (mixed pixels within the fire perimeter for Sentinel-2, 8.5%–13.5% vs. 31%–52% for Landsat OLI). In addition, dNBR had higher accuracy (higher coincidence values and less omission and commission) than dNDVI and dTST. In terms of fire severity degrees, dNBR (the best fire index candidate) derived from both satellites sensors were only capable of detecting the severe spots "severely-burned" with producer accuracy >70%. In fact, the dNBR-Sentinel-2/Landsat-8 overall accuracy and Kappa coefficient for classifying fire severity degree were less than 70% across the studied sites, except for Sentinel-dNBR in Dibbeen (72.5%). In conclusion, Sentinel-dNBR and Landsat promise to delimitate forest fire perimeters of small-scale (<25 ha) areas, but further remotely-sensed techniques are require (e.g., Landsat-Sentinel data fusion) to improve the fire severity-separability potential.

**Keywords:** remote sensing; thermal image; dNBR; NDVI; fire mapping; Kappa coefficient



## 1. Introduction

The forest is an essential variable in the ecological balance of the earth [1]. Forest fire intensity and extent has increased globally as the human imprint continues to intrude on natural areas and climate change effects increase the potential of extreme weather [2]. Fires burn millions of hectares of vegetation (including in the forest) every year, and increased fire extent has been reported in several global regions [3]. In fact, several large-scale forest fires have been erupted recently all the world for example, Australian wildlife fires which killed about a half-billion animals [4]. Forest fires are usually discovered after they spread across substantial spots, making them difficult to control [1]. In addition, forest fires are a destructive disaster, causing massive threats to human and ecological habitats and contributing to 30% of atmospheric carbon dioxide [1,4].

Forests fire intensity affects the survival and recovery of soil, fauna, and flora, and the pattern formation of forests, including succession and regeneration [5]. The results of forest fires are devastating and could last for several years, especially in arid and semi-arid regions [6]. The extent of drylands in the Mediterranean region has increased

extremely in recent decades and will continue to expand in the future because of the higher warming effects in this region compared to other regions [7]. In drylands, all types of natural hazards are expected to exist, however, climate hazards have been given greater attention in these highly dynamic environments [8]. The rural inhabitants of drylands (~one billion people) whose livelihoods are directly dependent on the physical environment encounter potential levels of risk from climate threats, some of which are expected to become more frequent and intense with climate change [8]. For example, the changing climate will have potential impact on future forest fires [9–11]. The shifts in forest community composition are associated with fire date, intensity, and consumption of upper soil layers, specifically organic soil horizons during recent fires [5]. Under exponential increase in earth temperature due to climate change, forest fire departments will encounter drier weather conditions that could push the present suppression capability beyond its tipping point, resulting in a significant increase in fire extent [9]. The modeling of wildfire risk in western Iran based on the integration analytic hierarchy process and geographic information system revealed that about 65% of the region was located in the high- and very high-risk zones [10]. Therefore, it is essential that the techniques of patrolling, detecting and fighting forest fires are available to forestry managers and responders. In this context, the development of integrated information systems from several sources and automated data processing chains is essential [2].

Fire detection in forests has become a major concern and a challenge due to its severity and extent [4]. The detection of burned area and the predictive risk of forest fires are crucial to preventing further damage, managing burned areas, and developing early forest fire warning systems [12–14]. A large number of applied and theoretical research studies have been conducted to detect forest fires early on [6,11,12]. Concurrently, many biological and environmental studies were also applied on fauna and flora that live in the forest, especially on the endangered species [15]. The remote sensing approach has been used widely to detect vegetation cover extent, density, and health of forests [11,16,17]. Remote sensing-based estimation is a reliable and economic technique for detecting vegetation status over large areas [12]. This technique can detect larger spatial extent of vegetation faster and with a lower cost than ground measurements [18]. Spatial and spectral data from remote sensors such as RapidEye, Sentinel-2, Landsat, PlanetScope imagery, and Google Earth Engine can potentially detect vegetation change across the year [13,19]. Landsat series [Landsat 8 Operational Land Imager (OLI), Landsat 7 Enhanced Thematic Mapper Plus (ETM+), and Landsat 5 Thematic Mapper (TM)] provide imagery with an outstanding optical resolution for land cover [20–23]. Landsat sensors have a moderate revisit period (16-day), a spatial resolution of 30 m, and a spectrum from visible to short wave near infrared (SWIR). In addition, thermal infrared sensor (TIRS) on Landsat platform can provide land surface temperature images which can then be used to detect forest fires. The Landsat series is still active to this day, making Landsat (launched in 1972) the longest continuous Earth imaging program in history [20]. Long-term series (more than 50 years) of freely Landsat images coupled with ground surveys have been considered reliable methods to quantify vegetation cover change, including burned areas [11,24]. Interestingly, Sentinel-2A sensors from the European Union Copernicus program provide extraordinary images since 2015 at spatial resolution ranging from 10 to 60 m, a revisit time of 5 days, and spectral resolution of 13 bands [25]. Remote sensing indices such as NDVI and NBR are derived from Sentinel-2 and Landsat sensors and can be used to identify forest fires areas and estimate burn severity [11,26]. For example, NDVI is normally used to evaluate the extent of a burned area while NBR is used to estimate the burn severity [26].

In arid and semi-arid regions including Jordan, the forests area is limited, heterogeneous and fragmented. The fire incidences in those dry lands have become more frequent and severe recently. Jordan is Mediterranean country, has long hot summers, and relatively short cold winters. The total forest area in Jordan is about 1%; dense forests 398 km$^2$, spares forest, 394 km$^2$ [27]. The most popular genera are *Quercus* and *Pinus* (specifically, *Pinus halepensis* Mill.). Forest fires due to hot dry climate and illegal logging are the main threats

to Jordanian forests. In 2021, more than 200 forest fire were reported by the Forestry Department in Jordan. Several research studies revealed a potential use of remotely-sensed data from moderate satellite sensor images such as, Sentinel-2 and Landsat series to detect fire burned area perimeter and severity level, especially when burned area is large (more than 100 ha) [11,12,26]. However, limited study have been assessed the usefulness of using these moderate spatial resolution sensors (Sentinel-2, 10–20 m; Landsat, 30 m) to precisely detect forest fire extent and severity at small-scale areas (less than 25 ha) in arid and semi-arid environment such as, Jordan. The objective of this study was to assess the use of Sentinel-2 and Landsat sensors data to detect small-extent (<25 ha) forest fires perimeter and severity in arid environments.

## 2. Materials and Methods

### 2.1. Study Sites

Three forest fires occurred in Northern Jordan between 2003 and 2020 were assessed (Figure 1). The first site was in Ajloun (fire date: October 2020; total burned area, 23 ha), the second site was at Sakeb, Jerash (Latroon Mountain, fire date: August 2003; total burned area, 15 ha), and the third site was at Dibbeen Natural Reserve (Aqra' Mountain, fire date: June 2016, total burned area 10.5 ha). The dominant tree species in Sakeb and Dibbeen Natural Reserve sites (more than 60%) is *Pinus halepensis*. In Ajloun site, the main species is *Quercus coccifera* L. (more than 90%). Both plant species are native to Jordanian lands [28]. In fact, Dibbeen Natural Reserve has the southernmost native *Pinus halepensis* forest in the world and the last remaining stand of old pine forest in Jordan [28]. Although the fires severity degree ranged from low to severe across sites, 40%–60% of the burned sites exposed to severe fire and consumed most of the trees.

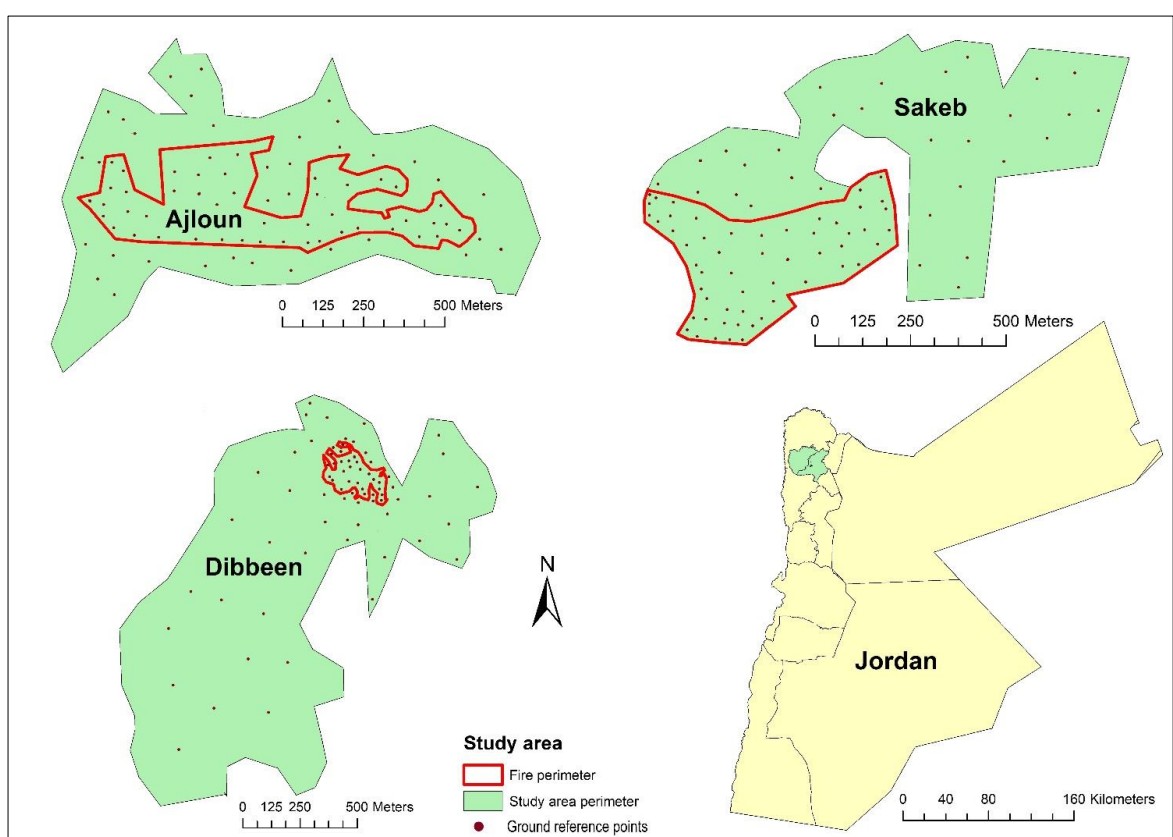

**Figure 1.** Location of the Ajloun, Dibbeen and Sakeb forest fire sites, Northern Jordan.

*2.2. Landsat and Sentinel-2 Sensors Data (Acquisition and Pre-Processing)*

Pre- and post-fire satellite sensors images (one month interval, two images) from Landsat-8 (OLI and TIRS) and Sentinel-2 (Multispectral Instrument, MSI) were used to detect vegetation cover change in Ajloun and Dibbeen sites after fire incidence. In addition, two images from Landsat-7 sensors (ETM+ and TIR) before and after fire (July and September 2003) were used to detect changes in vegetation at Sakeb forest fires site (Landsat-8 and Sentinel-2 were not lunched yet). Landsat OLI, ETM+ and Sentinel-2 (MSI) level 2 images were used. These images are atmospherically and radiometrically corrected [11,29] and are available for free from EarthExplorer website. Image preprocessing and calculation of vegetation indices were conducted using Environment for Visualizing Images (ENVI) 5.0 (Research Systems, Boulder, CO, USA). Landsat thermal images were processed following the procedures of Chander et al. [30]. Normalized Burn Ratio (NBR, Equation (1)), Normalized Different Vegetation Index (NDVI, Equation (2)) and thermal surface temperature (TST) images were used to calculate the differenced NBR (dNBR, Equation (3)), NDVI (dNDVI, Equation (4)) and (dTST, Equation (5)), which estimated the burned area extent and severity (Table 1) [31].

$$NBR = \frac{(NIR - SWIR)}{(NIR + SWIR)} \tag{1}$$

$$NDVI = \frac{(NIR - R)}{(NIR + R)} \tag{2}$$

$$dNBR = NBR \text{ prefire} - NBR \text{ postfire} \tag{3}$$

$$dNDVI = NDVI \text{ prefire} - NDVI \text{ postfire} \tag{4}$$

$$dTST = TST \text{ prefire} - TST \text{ postfire} \tag{5}$$

**Table 1.** Differenced Normalized Burn Ratio (dNBR) [31], Normalized Different Vegetation Index (dNDVI) and Thermal Surface Temperature (dTST).

| Index | Index Range | Fire Severity Level |
|---|---|---|
| dNBR | −0.5–0.1 | Un-burned |
|  | 0.1–0.25 | Low |
|  | 0.25–0.45 | Moderate |
|  | dNBR > 0.45 | High |
| dNDVI | −1.0–0.1 | Un-burned vegetation/Bare soil |
|  | 0.1–0.2 | Low |
|  | 0.2–0.3 | Moderately-burned |
|  | 0.3–0.4 | Moderate-severely burned |
|  | dNDVI > 0.4 | Severely-Burned |
| dTST | $0.0\,°C \leq dTST \leq$ mean temperature of the thermal image within the study area | Un-burned |
|  | mean temperature of the thermal image for the study area < dTST | Burned |

*2.3. Ground Reference Data and Accuracy Assessment of Remote Sensing Data*

Reference data for burned areas perimeter and severity were mapped using ground survey one week after the fire incidence. A GPS points represent the perimeter of the fire as well as the severity level (low, moderate and high) were also identified inside each forest fire site by forestry experts. Then, the burnet area perimeter and severity degrees were mapped using ArcGIS ArcMap (Version 10.2 for Windows; ESRI, Redlands, CA, USA) software.

Fire perimeter delimitation for the tested remotely sensed indices (dNBR, dNDVI, dTST) was determined following the procedures of Llorens et al. [11]. The spectral indices and the reference perimeters were compared to determine the fire perimeter. The spectral indices (dNBR, dNDVI, dTST) were first classed using the unburned and burned threshold in Table 1; dNBR = 0.1, dNDVI = 0.2, dTST = mean temperature of the thermal image within

the study area. To determine the accuracy of pixel values, the area of coincidence, omission and commission equations suggested by Llorens et al. [11] and Amos et al. [32] were used:

$$\text{Area of coincidence (\%)} = \frac{\text{Coincidence Pixels}}{(\text{Coincidence Pixels} + \text{Omission Pixels})} \times 100 \qquad (6)$$

$$\text{Area of omission (\%)} = \frac{\text{Omission Pixels}}{(\text{Coincidence Pixels} + \text{Omission Pixels})} \times 100 \qquad (7)$$

$$\text{Area of commission (\%)} = \frac{\text{Commission Pixels}}{(\text{Coincidence Pixels} + \text{Commission Pixels})} \times 100 \qquad (8)$$

where area of coincidence is when the spectral indices (dNBR, dNDVI, dTST) correctly delineation the burned area compared to the ground reference points. The area of commission represents the pixels where the spectral index senses as no fire (values below thresholds) but the ground reference data detects as fire. The area of omission counts the pixels, where spectral index senses as fire, but the ground reference point detects as no fire. The total number of validation points for Ajloun site was 81, Dibbeen 70 and Sakeb 82. The best spectral index candidate for delineating the fire perimeter (from both satellites sensors) was further explored to determine its potential for identifying the burn severity levels. A confusion (error) matrix was developed for each site, in order to assess the User's, Producer's and Overall accuracy as well as Kappa coefficient [29,33].

## 3. Results and Discussion

### 3.1. Forest Fire Burned Area (Fire Perimeter Delimitation)

The forest fire perimeter delineation analysis was applied following Equations (6)–(8) that suggested by Llorens et al. [11] and Amos et al. [32]. The separability assessment of the two sensors' vegetation indices is presented in Table 2 and Figure 2. In each differenced index (dNBR, dNDVI, dTST), the highest values of coincidence and the lowest values of omission and commission represent the best candidate to discriminate between burned and unburned areas [11]. In this study, Sentinel-2 had higher coincidence area and lower omission and commission area than Landsat in both sites and indices (Table 2). In the first site (Ajloun), the coincidence area for Sentinel-2 derived indices ranged from 78.5% (dNDVI) to 81.3% (dNBR) while the percentage of coincidence for Landsat were between 62.0% and 64.4%. Similarly, the area of coincidence in the second site (Dibbeen forest fires) ranged from 82.5% to 84.1% for Sentinel-2 and about 78% for Landsat-derived spectral indices (Table 2). Sentinel-2 (MSI) data can be recommended as a key Earth observation data source in forest resources assessment and monitoring [34]. Howe et al. [35] assess the accuracy of Sentinel- and Landsat-derived burn indices for 26 fires that burned between 2016 and 2019 in western North America. They concluded that burn severity mapping could significantly benefit from the integration of Sentinel imagery to Landsat-forest surveillance data by increasing imagery availability (image every 5 days), and that Sentinel's higher spatial resolution can improve opportunities for inspecting finer-scale fire impact across ecosystems. The spatial analysis of Portuguese forest fires in 2016 using Landsat 8 (OLI) and Sentinel-2 (MSI) sensors data revealed that the difference in fire burned area extent between Landsat-NDVI and field data was 13.3% and for Sentinel 2-NDVI was less than 7.8% [26]. They attributed the higher accuracy in Sentinel-2 data (compared to OLI) to higher spatial resolution in MSI sensor [26]. Both satellite sensors (Landsat-8 OLI, and Sentinel-2 MSI) have similar bands (i.e., spectral resolution) in the red, NIR and SWIR spectral regions [12]. However, Sentinel-2 spatial resolution for those bands are 10 m for red and NIR and 20 for SWIR but the spatial resolution for Landsat-8 OLI spectral bands is 30 m. Therefore, the higher accuracy of delimiting fire perimeter in this study in Sentinel-2 indices (compared to Landsat-8 OLI) can be partially attributed to higher spatial resolution at Sentinel-2 MSI.

**Table 2.** Fire perimeter assessment (coincidence, omission and commission) of Landsat and Sentinel-2 differenced Normalized Burn Ratio (dNBR), Normalized Different Vegetation Index (dNDVI) and thermal surface temperature (dTST) for Ajloun, Dibbeen and Sakeb wildfires.

| | | Satellite | Vegetation Index | | |
|---|---|---|---|---|---|
| | | | dNBR | dNDVI | dTST |
| Ajloun | Area of coincidence (%) | Sentinel-2 | 81.3 | 78.5 | na * |
| | | Landsat-8 | 64.4 | 62.0 | 78.7 |
| | Area of omission (%) | Sentinel-2 | 21.5 | 11.9 | |
| | | Landsat-8 | 38.0 | 37.0 | 21.3 |
| | Area of commission (%) | Sentinel-2 | 9.0 | 19.2 | na |
| | | Landsat-8 | 14.5 | 18.5 | 9.2 |
| Dibbeen | Area of coincidence (%) | Sentinel-2 | 84.1 | 82.5 | na |
| | | Landsat-8 | 78.3 | 78.2 | 55.2 |
| | Area of omission (%) | Sentinel-2 | 15.9 | 17.5 | na |
| | | Landsat-8 | 21.7 | 21.8 | 44.8 |
| | Area of commission (%) | Sentinel-2 | 1.7 | 11.9 | na |
| | | Landsat-8 | 1.8 | 25.9 | 7.5 |
| Sakeb | Area of coincidence (%) | Sentinel-2 | - ** | - | - |
| | | Landsat-7 | 87.7 | 82.1 | 85.7 |
| | Area of omission (%) | Sentinel-2 | - | - | - |
| | | Landsat-7 | 12.3 | 17.9 | 14.3 |
| | Area of commission (%) | Sentinel-2 | - | - | - |
| | | Landsat-7 | 9.0 | 27.6 | 26.0 |

* na, not available. Sentinel-2 has no thermal bands; dTST cannot be derived. ** Launch date for Sentinel-2 is June 2015; no images are available for Sakeb forest fires occurred in 2003.

During the study period, three differenced vegetation indices were assessed; the dNBR (NIR and SWIR), dNDVI (red and NIR) and dTST (thermal band) which inclusively available in Landsat-7 (TIR) and 8 (TIRS) (Table 2, Figure 2). The dNBR index had higher coincidence area percentage than dNDVI across the studied sites and satellite sensors. In addition, the same index (dNBR) had lower omission and commission values at both satellite sensors and across the sites, except for omission area percentage at Sentinel-2 in Ajloun site (Table 2). Interestingly, this study revealed that the use of dNBR derived from Sentinel-2 MSI sensor consistently had higher coincidence area than dTST derived from Landsat-thermal sensors (TIR and TIRS). In Marmara region of Turkey (large-extent forest), the assessment of NDVI, NBR, and enhanced vegetation index (EVI) using Landsat-8 sensor data revealed that the overall accuracy of NDVI, and NBR in deciduous forests was around 85% and 78.80% for EVI, while in coniferous forests, the overall accuracy was between 87% to 88% for all tested vegetation indices [36]. Water-sensitive vegetation indices (e.g., NBR) which include SWIR are more sensitive for detecting forest disturbances (specifically forest fires) while chlorophyll-sensitive indices (NDVI) represents lower accuracy [36]. The best predictive image bands (Sentinel-2, Landsat-8) for forest fires is normally the SWIR bands [6,34]. These bands (SWIR) exhibit a gradual increase in reflectance over time as a result of the decrease in water absorption in burned areas, in contrast to NIR and green bands between pre- and post-fire (the absence of vegetation cover) imagery [11]. Therefore, the index that comprises data from the NIR and SWIR spectrum regions is the candidate that should be used to discriminate between burned and unburned areas [11].

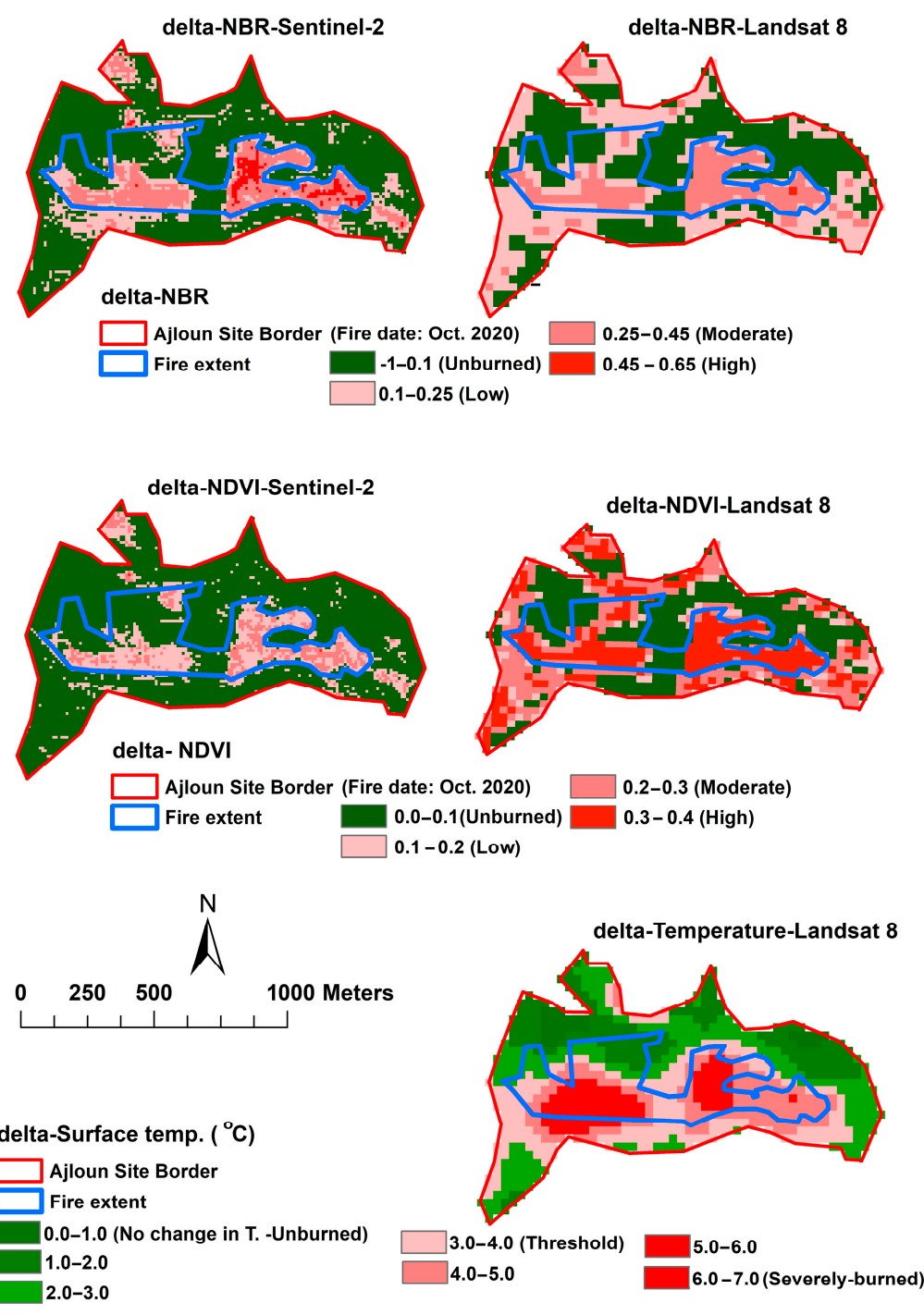

**Figure 2.** Landsat (OLI/ETM+) and Sentinel-2 (MSI) differenced Normalized Burn Ratio (dNBR), Normalized Different Vegetation Index (dNDVI) and thermal surface temperature (dTST) for Ajloun, Dibbeen and Sakeb wildfires, Northern Jordan.

The use of remote sensing technique for forest fire detection is not new. Several research results revealed a potential and accurate detection of "large-scale" forest fire perimeter using Landsat and Sentinel-2 sensors data [11,37,38]. However, limited studies we aware of have been focused on small extent forest fires (less than 25 ha), especially in arid environments [6]. In this study, the ground measurements of forests fire showed that the total fire area in Ajloun forest site was 23 ha; Dibbeen was 10 ha and Sakeb burned area extent was about 15 ha. The accuracy of dNBR (best candidate) for differentiating

fire perimeter ranged from 64%–81% at Ajloun, 78%-84% at Dibbeen and 87% at Sakeb. Compared to large-extent forest fire studied, the accuracy of this study in delineating fire perimeter for small-extent forest fires is about 10%–20% lower. When investigators attempt to extract information from satellite images which is smaller than the size of the pixel (tree often smaller than either the 10 m resolution of the Sentinel-2 MSI or the 30 m of the Landsat OLI), it lead to the mixed pixel problem; whether they contain (1) boundaries between two or more objects (road + vegetation, vegetation + bare soil, etc.) or (2) linear sub-pixel objects (a road within a dense vegetation) [39]. Therefore, the probability of mixed pixel problem in small irregular-shape burned area is expected to be higher than the large-scale regular-shape (e.g., rounded or squared plots) forest fires. In this study, the percentage of total pixels located within the perimeter of burned area at Ajloun was between 13.5% (Sentinel-2) and 52.2% (Landsat OLI), Dibbeen ranged from 8.5% (Sentinel-2) and 31.3% (Landsat OLI) and for Sakeb was 34.1% (Landsat OLI) (Table 3). Given that the accuracy (%) of delimitation fire perimeter at Ajloun site was lower than Dibbeen and Sakeb while the percentage of pixels located within the perimeter of burned area in the same location (Ajloun) was extremely higher than the other tested sites, we believe that the mixed pixel problem significantly reduced the separability potential of remotely-sensed data.

**Table 3.** Total burned area (ground reference data), total number of pixels inside and within the perimeter burned area in Ajloun, Dibbeen and Sakeb forest fires, Northern Jordan. Satellite sensor data were from Landsat OLI (Ajloun, Dibbeen)/ETM+ (Sakeb) and Sentinel-2 MSI (Ajloun and Dibbeen).

| | Sensor | Site | | |
|---|---|---|---|---|
| | | Ajloun | Dibbeen | Sakeb |
| Total burned area (ha) | Ground reference | 23 | 10.5 | 15 |
| Total number of pixels inside the burned area | Landsat OLI/ETM+ | 255 | 115 | 167 |
| | Sentinel-2 | 2321 | 1018 | - * |
| Total number of pixels across the perimeter of burned area | Landsat OLI/ETM+ | 133 | 36 | 57 |
| | Sentinel-2 | 314 | 87 | - |
| Percentage of pixels located within the perimeter of burned area (%) | Landsat OLI/ETM+ | 52.2 | 31.3 | 34.1 |
| | Sentinel-2 | 13.5 | 8.5 | - |

* Launch date for Sentinel-2 is June 2015; no images are available for Sakeb forest fires occurred in 2003.

### 3.2. Assessment of Burn Severity Levels

Forest is crucial resource that protect ecological balance on Earth [40]. Severe forest fire negatively affect regional weather patterns, the presence of endangered flora and fauna, global warming as well as human's safety and their financial resources [1]. Forest fire detection and suppression efforts have been increased recently in an attempt to mitigate its impact. According to statistics, fires cause severe hazard to environment, industries, and human and animal life around the world [4]. However, forest fire extent and severity detection has become a major worry and a difficult task. In this study, a deeper analysis of post-fire has been performed to assess if the satellite images data are able to discriminate between burn severity levels. The assessment process included, confusion matrix, Kappa coefficient and accuracy assessment for the best index candidate in delimitating the fire perimeter (dNBR) for both Sentinel-2 (Table 4) and Landsat OLI/ETM+ (Table 5).

**Table 4.** Confusion matrix and accuracy assessment of burn severity levels derived using Sentinel-2 differenced Normalized Burn Ratio (dNBR), for Ajloun (2020) and Dibbeen (2016) wildfires.

| | Reference Data | | | | Total |
|---|---|---|---|---|---|
| | Unburned | Low | Moderate | High | |
| Ajloun site | | | | | |
| Unburned | 30 | 7 | 4 | 0 | 41 |
| Low | 5 | 3 | 1 | 0 | 9 |
| Moderate | 0 | 1 | 8 | 1 | 10 |
| High | 0 | 0 | 4 | 12 | 16 |
| Total | 35 | 11 | 17 | 13 | 76 |
| Producer Accuracy (%) | 85.7 | 27.3 | 47.1 | 92.3 | |
| User Accuracy (%) | 73.2 | 33.3 | 80.0 | 75.0 | |
| Overall Accuracy (%) | | | | | 64.2 |
| Kappa coefficient (%) | | | | | 54.8 |
| Dibbeen site | | | | | |
| Unburned | 56 | 8 | 0 | 0 | 64 |
| Low | 1 | 4 | 1 | 0 | 6 |
| Moderate | 0 | 1 | 4 | 0 | 5 |
| High | 0 | 0 | 2 | 14 | 16 |
| Total | 57 | 13 | 7 | 14 | 91 |
| Producer Accuracy (%) | 98.2 | 30.8 | 57.1 | 100 | |
| User Accuracy (%) | 87.5 | 66.7 | 80.0 | 87.5 | |
| Overall Accuracy (%) | | | | | 76.0 |
| Kappa coefficient (%) | | | | | 72.5 |

The Sentinel-dNBR results showed that the index was able to differentiate between unburned and severely-burned regions (producer and user accuracy > 70%) (Table 4). However, the ability of Sentinel-dNBR to identify the low and moderate burned area was low (<70%). In addition, the overall accuracy for Sentinel-dNBR at Ajloun was 64.2% and 76.0% for Dibbeen forest fires. For dNBR-Landsat, the overall accuracy for Ajloun was 46.7%, Dibbeen was 63.5% and Sakeb was 60.1%. The producer and user accuracy were consistently higher than 70% at unburned and severely burned classes in Dibbeen and Sakeb sites (Table 5). Llorens et al. [11] proposed a methodology to estimate the extent and burn severity of forest fires occurred in 2017 in Spain and Portugal using Sentinel 2 images (10 and 20 m). The comparison with the European Forest Fire Information System (EFFIS) revealed that severity levels from Sentinel-2 and EFFIS were highly correlated the Separability index higher >1 and Kappa statistic > 69% [11]. Quintano et al. [37] concluded that the overall accuracy and Kappa coefficient ≥ 70% is adequate level for be used by forest managers. In this study, the overall accuracy and Kappa coefficient were less than 70% across the studied sites (Ajloun, Dibbeen and Sakeb) and in both Satellite sensors, except for Sentinel-dNBR at Dibbeen (72.5%). The analyses of Landsat-8, Sentinel-2, and Terra satellites sensor images for Brazil and Bolivia forest fires in 2020 showed an overall accuracy of 90% but kappa coefficient value was 0.65 [41]. Mashhadi and Alganci [36] found that all vegetation indices underestimated the deforested area. Overall, in small-scale forest fires (<25 ha), the use of dNBR is recommended only to delineate the forest fire perimeter and "severely-burned" class.

**Table 5.** Confusion matrix and accuracy assessment of burn severity levels derived using Landsat differenced Normalized Burn Ratio (dNBR), for Ajloun (2020), Dibbeen (2016) and Sakeb (2003) wildfires.

| | Reference Data | | | | Total |
|---|---|---|---|---|---|
| | Unburned | Low | Moderate | High | |
| Ajloun site | | | | | |
| Unburned | 20 | 19 | 4 | 0 | 43 |
| Low | 4 | 3 | 1 | 0 | 8 |
| Moderate | 0 | 1 | 9 | 1 | 11 |
| High | 0 | 0 | 14 | 2 | 16 |
| Total | 24 | 23 | 28 | 3 | 78 |
| Producer Accuracy (%) | 83.3 | 13.0 | 32.1 | 66.7 | |
| User Accuracy (%) | 46.5 | 37.5 | 81.8 | 12.5 | |
| Overall Accuracy (%) | | | | | 46.7 |
| Kappa coefficient (%) | | | | | 23.9 |
| Dibbeen site | | | | | |
| Unburned | 50 | 14 | 0 | 0 | 64 |
| Low | 2 | 2 | 0 | 0 | 4 |
| Moderate | 0 | 2 | 3 | 0 | 5 |
| High | 0 | 1 | 4 | 12 | 17 |
| Total | 52 | 19 | 7 | 12 | 90 |
| Producer Accuracy (%) | 96.2 | 10.5 | 42.9 | 100 | |
| User Accuracy (%) | 78.1 | 50.0 | 60.0 | 70.6 | |
| Overall Accuracy (%) | | | | | 63.5 |
| Kappa coefficient (%) | | | | | 53.6 |
| Sakeb site | | | | | |
| Unburned | 54 | 6 | 0 | 0 | 60 |
| Low | 5 | 1 | 0 | 0 | 6 |
| Moderate | 0 | 2 | 3 | 0 | 5 |
| High | 0 | 0 | 5 | 14 | 19 |
| Total | 59 | 9 | 8 | 14 | 90 |
| Producer Accuracy (%) | 91.5 | 11.1 | 37.5 | 100 | |
| User Accuracy (%) | 90.0 | 16.7 | 60.0 | 73.7 | |
| Overall Accuracy (%) | | | | | 60.1 |
| Kappa coefficient (%) | | | | | 61.4 |

## 4. Conclusions

We explored the potential use of Sentinel-2 (MSI), Landsat (OLI, ETM+ TIR, TIRS) for detecting small-scaled (<25 ha) forest fire perimeter and severity in Northern Jordan. Differenced NBR (dNBR) derived from both satellite sensors had higher coincidence values and less omission and commission than dNDVI and dTST. Compared to Landsat the use of Sentinel-2 sensor spectral reflectance data (specifically, dNBR) had increased the coincidence area by 7%–26% and reduced the area of omission by 28%–43% and area of commission by 6%–38%. This is attributed partially to higher Sentinel-2 spatial resolution. However, the separability potential of fire perimeter ranged from 81%–84% for Sentinel-dNBR and from 64%–88% for Landsat-dNBR. This is because the percentage of total pixels located within the perimeter of burned area was high (Sentinel 8.5%–13.5%; Landsat 31.3%–52.2%) and resulted in a mixed pixel problem. In term of ability of Sentinel-2 and Landsat sensor indices to classify burn severity, both satellite sources were able to detect only the "severely-burned" area (Producer accuracy > 70%). However, both satellite sensors burn index (dNBR) failed to detect low or moderate burned area properly (Producer and User accuracies < 70%). In addition, the overall accuracy and Kappa coefficient were less than 70% across the studied sites (Ajloun, Dibbeen and Sakeb) and in both Satellite sensors, except one site. Overall, in "small-scale" burned area (<25 ha), Sentinel-2 and Landsat derived dNBR were able to delineate the forest fire perimeter and identify the "severely-burned" area. However, the other related forest factors, such as terrain slope, forest species, and underlying surface might affect the accuracy level in other regions. To improve the fire severity classification

potential for these moderate space-borne satellites, Landsat-Sentinel data fusion could be a viable alternative to reduce relative errors (e.g., mixed pixels) and increase accuracy.

**Author Contributions:** Conceptualization, Y.A.O. and B.Q.; methodology, Y.A.O., H.A.A. and B.A.Q.; software, Y.A.O.; validation, Y.A.O., H.A.A. and B.A.Q.; formal analysis, Y.A.O.; investigation, B.Q. and Y.A.O.; data curation, Y.A.O.; writing—original draft preparation, B.Q., Y.A.O. and M.A.-A.; writing—review and editing, Y.A.O.; supervision, Y.A.O.; project administration, Y.A.O. and B.Q. All authors have read and agreed to the published version of the manuscript.

**Funding:** This research received no external funding.

**Data Availability Statement:** Not applicable.

**Acknowledgments:** The authors are grateful to the Deanship of Scientific Research at University of Jordan for partially-funded this project.

**Conflicts of Interest:** The authors declare no conflict of interest.

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
