# Peer review of "Assessment of Small-Extent Forest Fires in Semi-Arid Environment in Jordan Using Sentinel-2 and Landsat Sensors Data"

_forests, doi:10.3390/f14010041_

Round 1

Reviewer 1 Report

This paper evaluated the separability potential of Sentinel-2A and Landsat derived indices for detecting small-extent forest fires area and severity degrees in arid environments.

1. There are several relevant references that provide sufficient background must be included in introduction.

2. The title should be changed to emphasize the features of small-extent and arid environment

3. It seems impossible to improve accuracy by fusing the Landsat-Sentinel data to reduce relative errors (e.g. mixed pixels). please confirm it.

4. As the results derived from remote sensing image may contain many uncertainties, the referenced data with total 76 and 91 samples for accuracy evaluation are inadequate to generate stable results. Moreover, it is suggested to plot the spatial distribution of the samples, due to it also influences the final results.

5. For my part, the main conclusion of the paper (in line 324-326) is preliminary and its generalization is weak, due to the authors do not analyze other related factors in experiments, such as terrain slope, forest species, and underlying surface et al.  

Author Response

Thank you for the constructive and thoughtful comments. As outlined below and in the manuscript text, we have revised it following your comments and suggestions. We hope that you will now find the revised version of the manuscript suitable for publication.

Reviewer comments

This paper evaluated the separability potential of Sentinel-2A and Landsat derived indices for detecting small-extent forest fires area and severity degrees in arid environments.

  1. There are several relevant references that provide sufficient background must be included in introduction.

Answer: We agree. The introduction revise and new references added to provide sufficient background.   

  1. The title should be changed to emphasize the features of small-extent and arid environment

Answer: We agree. The title revised as follow “Assessment of small-extent forest fires in semi-arid environment in Jordan using Sentinel-2 and Landsat sensors data”

  1. It seems impossible to improve accuracy by fusing the Landsat-Sentinel data to reduce relative errors (e.g. mixed pixels). Please confirm it.

Answer: In our study we used Landsat and sentinel data seperatly. We did not combine (fuse) the data of both satellites. A recent study showed that the fusion of both data improved the accuracy for dense large-scale forests. Therefore, we recommended this approach for the future study in our conclusions. Overall, we cannot confirm that the accuracy will not be improve without testing this hypothesis in the future study. 

  1. As the results derived from remote sensing image may contain many uncertainties, the referenced data with total 76 and 91 samples for accuracy evaluation are inadequate to generate stable results. Moreover, it is suggested to plot the spatial distribution of the samples, due to it also influences the final results.

Answer: We understand the reviewer concerns of that the training points are not adequate. However, in this study the size of fire extent for the sites is less than 25 ha which mean we have more than 2 training points within the fire extent and at least 1 outside. We believe that this number of reference is adequate when compared to previous studies. 

In term of spatial distribution of samples, we totally agree with the reviewers. The reference points added to revised maps.     

  1. For my part, the main conclusion of the paper (in line 324-326) is preliminary and its generalization is weak, due to the authors do not analyze other related factors in experiments, such as terrain slope, forest species, and underlying surface et al.

Answer: Good point. We revised the conclusion to highlight the reviewer suggestions. 

Reviewer 2 Report

The manuscript is an excellent work. The framework of forest fire assessment is also useful for investigating the effects of the forest fire in other watersheds and plan more extensive field studies to test predictions of hypotheses. The experiment was logically designed and very systematically implemented. The manuscript is also very clearly structured and well-written. Overall, I do only have a few comments the authors may want to consider:

- I am very keen to read the latest paper in this field, so I found some papers (be sure I am not one of the authors of the following appears), and authors can use to enrich the introduction and discussion sections  about the forest fire:

 ** https://doi.org/10.1007/s10661-022-10318-y

** https://doi.org/10.1109/JSTARS.2022.3225070

- L 70: Also add this citation: https://doi.org/10.3390/rs14091977
L 85: Add some statistics about the forest in Jordan.
L 102: Is this tree natural or the plantation? Add some overall information about forest stand attributes in each region (like average of DBH, DENSITY, etc).

Author Response

Thank you for the constructive and thoughtful comments. As outlined below and in the manuscript text, we have revised it following your comments and suggestions. We hope that you will now find the revised version of the manuscript suitable for publication.

Reviewer comments

The manuscript is an excellent work. The framework of forest fire assessment is also useful for investigating the effects of the forest fire in other watersheds and plan more extensive field studies to test predictions of hypotheses. The experiment was logically designed and very systematically implemented. The manuscript is also very clearly structured and well-written. Overall, I do only have a few comments the authors may want to consider:

- I am very keen to read the latest paper in this field, so I found some papers (be sure I am not one of the authors of the following appears), and authors can use to enrich the introduction and discussion sections about the forest fire:

 ** https://doi.org/10.1007/s10661-022-10318-y

Answer: The suggested reference added.

** https://doi.org/10.1109/JSTARS.2022.3225070

Answer: The suggested reference added.

- L 70: Also add this citation: https://doi.org/10.3390/rs14091977

Answer: The suggested reference added.

L 85: Add some statistics about the forest in Jordan.
Answer: Good point, the forest area and forest fire statistics in Jordan added to the revised version of the manuscript.  

L 102: Is this tree natural or the plantation? Add some overall information about forest stand attributes in each region (like average of DBH, DENSITY, etc).

Answer: We agree. The description of the forest trees in the studied sites added to the materials and methods section. 

Round 2

Reviewer 1 Report

Authors have replied most of my concerns, and the revised version has been improved.  The references are still insufficient, and more recently published relevant references can be cited. 

Author Response

Thank you for the constructive and thoughtful comments. As outlined below and in the manuscript text, we have revised it following your comments and suggestions. We hope that you will now find the revised version of the manuscript suitable for publication.

Reviewer comments

Authors have replied most of my concerns, and the revised version has been improved.  The references are still insufficient, and more recently published relevant references can be cited. 

Answer: We agree. The following recently published relevant references have been added to the revised version of the manuscript.

Astola, H.; Häme, T.; Sirro, L.; Molinier, M.; Kilpi, J. Comparison of Sentinel-2 and Landsat 8 imagery for forest variable prediction in boreal region. Remote Sens. Environ. 2019, 223, 257-273. https://doi.org/10.1016/j.rse.2019.01.019.

Howe, A.A.; Parks, S.A.; Harvey, B.J.; Saberi, S.J.; Lutz, J.A.; Yocom, L.L. Comparing Sentinel-2 and Landsat 8 for Burn Severity Mapping in Western North America. Remote Sens. 202214, 5249. https://doi.org/10.3390/rs14205249

Mashhadi, N.; Alganci, U. Evaluating BFAST Monitor algorithm in monitoring deforestation dynamics in coniferous and deciduous forests with Landsat time series: A case study on Marmara region, Turkey. ISPRS Int. J. Geo-Inf. 2022, 11, 573. https:// doi.org/10.3390/ ijgi11110573.

Negri, R.G.; Luz, A.E.O.; Frery, A.C.; Casaca, W. Mapping burned areas with multitemporal– multispectral data and probabilistic unsupervised learning. Remote Sens. 2022, 14, 5413. https://doi.org/ 10.3390/rs14215413.